# Genome-Wide Identification and Analysis of the MADS-Box Gene Family in *Theobroma cacao*

**DOI:** 10.3390/genes12111799

**Published:** 2021-11-15

**Authors:** Qianqian Zhang, Sijia Hou, Zhenmei Sun, Jing Chen, Jianqiao Meng, Dan Liang, Rongling Wu, Yunqian Guo

**Affiliations:** 1Center for Computational Biology, College of Biological Sciences and Technology, Beijing Forestry University, Beijing 100083, China; awayzqq@163.com (Q.Z.); hsj381552790@163.com (S.H.); shixiaohuaa0201@163.com (J.C.); mjq990521@163.com (J.M.); liangdanyx2014@163.com (D.L.); rwu@bjfu.edu.cn (R.W.); 2Institute of Marine Materials Science and Engineering, College of Ocean Science and Engineering, Shanghai Maritime University, Shanghai 201306, China; sun1120817625@163.com

**Keywords:** MADS-box transcription factors, *Theobroma cacao*, bioinformatics analysis, genome-wide characterization, gene family

## Abstract

The MADS-box family gene is a class of transcription factors that have been extensively studied and involved in several plant growth and development processes, especially in floral organ specificity, flowering time and initiation and fruit development. In this study, we identified 69 candidate MADS-box genes and clustered these genes into five subgroups (Mα: 11; Mβ: 2; Mγ: 14; Mδ: 9; MIKC: 32) based on their phylogenetical relationships with *Arabidopsis*. Most *TcMADS* genes within the same subgroup showed a similar gene structure and highly conserved motifs. Chromosomal distribution analysis revealed that all the *TcMADS* genes were evenly distributed in 10 chromosomes. Additionally, the cis-acting elements of promoter, physicochemical properties and subcellular localization were also analyzed. This study provides a comprehensive analysis of MADS-box genes in *Theobroma cacao* and lays the foundation for further functional research.

## 1. Introduction

MADS-box genes encode eukaryotic transcription factors that play a prominent role in plant development processes. MADS-box proteins contain a highly conserved DNA-binding MADS-domain of approximately 50–60 amino acids in length in their N-terminal region, and this domain could be involved in recognizing and binding the CArG motif of their target gene [1]. The name itself is given by the initials of the four first-discovered transcription factors in this family, which are *MCMI* in *Saccharomyces cerevisiae* [2], *AGAMOUS* in *Arabidopsis thaliana* [3], *DEFICENS* in *Antirrhinum majus* [4] and *SRF4* in *Homo sapiens* [5]. Based on protein domain structure, the MADS-box genes are divided into two categories: type I and type II. The type I MADS-box genes can be further classified into Mα, Mβ, Mγ, Mδ subclasses. Type II lineage, also known as MIKC type, has a special MIKC structure, which is composed of an N-terminal MADS domain, the I (intervening) and K (keratin-like) regions and a variable C-terminal transcriptional activation domain [6]. Type MIKC were further divided into two subgroups, MIKC^C^ and MIKC*, according to their MIKC structural features [7].

The MADS-box gene family is known to have functions in many significant physiological and developmental processes, such as the regulation of floral organ specificity [3,4], control of flowering signals and initiation [8,9], fruit development [10], meristem identify specification [11], and seed development [12]. For example, Wheat *VERNALIZATION1* (*VRN1*) is a key regulator of flowering time and floral meristem determination [13] The MADS-box gene *FLOWERING LOCUS C* (*FLC*) controls the vernalization pathway in *Arabidopsis* [14]. Apple *MdDAM1* plays a role in bud dormancy and growth cessation in autumn [15]. Although MADS-box genes are well-known for their roles in the flower developmental process and participating in the classical ABC flower development model, some of them have been validated to function on root and leaf morphogenesis [16,17]. To date, the MADS-box proteins have been characterized in various kinds of plants, including *Arabidopsis* [18], *Populus trichocarpa* [19], *pineapple* [20], *Saccharum spontaneum* [21], *Erigeron breviscapus* [22]. However, little is known regarding the MADS-box gene family in *Theobroma cacao*. 

*Theobroma cacao* is an economically important tropical tree, native to South America, which is planted in large quantities for its fruits (cacao pods), where its beans were used as the raw material for making chocolate, coco butter, cosmetics and confectionery [23]. Additionally, some studies have proposed that an ingredient found in coco might exert cardiovascular benefits [24]. Research into the sequencing and assembling genome of *Theobroma cacao* was carried out in 2010 [25], leading to the genome-wide identification and analysis of important gene families such as the NAC domain transcription factor family [26], WRKY transcription factor family [27], and GPX family [28]. The metabolome and transcriptome profiling of the *Theobroma cacao* pods was completed [29]. In this scenario, we conducted a bioinformatics analysis of MADS-box members of *Theobroma cacao* at the gene level. We identified 69 MADS-box gene members, investigated their phylogenetic relationship, classified them, and analyzed gene structures, motifs, and chromosome location. Moreover, subcellular localization and cis-acting elements were also performed. Our results may provide a basis for further functional studies of coco tree genes and references for subsequent research into molecular mechanisms.

## 2. Materials and Methods

### 2.1. Identification of MADS-Box Genes in Theobroma cacao

*Theobroma cacao* genome sequences and annotation files were provided by Ensembl Plants (http://plants.ensembl.org/index.html, accessed on 16 April 2021. The hidden Markov model (HMM) profile of the MADS-domain was retrieved from the Pfam database (release 34.0; http://pfam.xfam.org/, accessed on 16 April 2021) with the accession number ‘PF00319’ [30].

MADS-box proteins in *Theobroma cacao* were searched using the following two approaches. First, the downloaded HMM profile was employed using the HMMER v3.3.2 program to search proteins containing the MADS-domain. Secondly, to avoid missing candidates, we constructed a new HMM model with proteins with e-value < 1 × 10^−20^, and ClustalW (version 2.1) was used for multiple sequence alignments [31]. The new model was used to search all *Theobroma cacao* protein sequences using HMMER (version 3.3.2), with a cut-off e-value of 0.05. Additionally, the predicted proteins were invalidated by conducting protein domain searches on the SMART program (http://smart.embl-heidelberg.de/, accessed on 19 June 2021) and NCBI Conserved Domain Search (https://www.ncbi.nlm.nih.gov/Structure/cdd/wrpsb.cgi, accessed on 19 June 2021) to confirm the presence of the MADS-domain in all candidate proteins.

### 2.2. Phylogenetic Analysis and Classification of MADS-Box Genes

To understand the phylogenetic relationship and to classify the MADS-box genes, a rooted neighbor-joining (NJ) phylogenetic tree for *Theobroma cacao* (*TcMADS*) and *Arabidopsis* MADS-box proteins was constructed using MEGA X software (version 10.2.2) [32]. The *TcMADS* gene family was classified according to their phylogenetic relations with corresponding *Arabidopsis* MADS-box members. *Arabidopsis* MADS-box protein sequences were downloaded from TAIR (https://www.arabidopsis.org/, accessed on 24 July 2021) with the accession numbers reported by Parenicová et al [18]. All protein sequences were aligned by Muscle with the default parameters [33]. The Neighbor-Joining method was used, with the following parameters: 1000 replications for bootstrap method, Poission model, Pairwise deletion. Additionally, an individual phylogenetic tree of *TcMADS* genes was built with the same method and beautified by ggtree [34].

### 2.3. Conserved Motif and Gene Structure Analysis

Online program MEME (https://meme-suite.org/meme/tools/meme, accessed on 29 July 2021) was applied to analyze the conserved motifs in the MADS-box protein with the following settings: maximum number of motifs 10, minimum motif width 6, maximum motif width 50, number of repetitions any [35]. The intron–exon structure information was contained in the *Theobroma cacao* gtf file downloaded from Ensembl Plants. Conserved motif and gene structure were both visualized by TBtools software (version 0.665).

The online tools ProtParam (https://web.expasy.org/protparam/, accessed on 10 August 2021) and Compute pI/Mw (https://web.expasy.org/compute_pi/, accessed on 10 August 2021) was employed to analyze physicochemical properties including theoretical isoelectric points (PI), average molecular weight (MW), instability index and aliphatic index. Number of amino acids (aa) and open reading frame (ORF) lengths were both found with the ORFfinder website (https://www.ncbi.nlm.nih.gov/orffinder/, accessed on 11 August 2021). The BUSCA program (https://busca.biocomp.unibo.it/, accessed on 4 August 2021) was used to predict TcMADS proteins’ subcellular localization (SL).

### 2.4. Chromosomal Localization and Gene Duplication

The locational information on the chromosomes and chromosome length of *TcMADS* genes was acquired from Ensembl Plant. All identified genes were mapped to 10 chromosomes with MG2C (http://mg2c.iask.in/mg2c_v2.1/, accessed on 25 June 2021) according to their chromosomal positions and relative distance. *TcMADS* gene potential duplication was confirmed based on major criteria as follows: (a) sequence alignment length cover > 75% of longer sequence, and (b) the similarity of the aligned region > 75% [36]. Bio-Linux was used to screened tandem repeat sequences. The TcMADS protein sequences were aligned by MAFFT (version 7.481), and then multiple protein alignment were confirmed and the corresponding DNA sequences were sorted into codon alignments [37], which were used to calculate the Ka/Ks ratios using KaKs calculator Toolbox 2.0 (version 2.0).

### 2.5. Analysis of Cis-Acting Element in MADS-Box Genes’ Promoters

The upstream sequences (2 kb) of *TcMADS* genes’ CDS were retrieved from the *Theobroma cacao* genome by TBtools software according to gene ID, and then submitted to PlantCARE (http://bioinformatics.psb.ugent.be/webtools/plantcare/html/, accessed on 5 August 2021) to identify four cis-acting elements, including light-responsive elements, wound-responsive elements, gibberellin-responsive elements, and auxin-responsive elements, after filtering and screening. The variety and quantity of cis-acting elements upstream each gene was found with TBtools.

## 3. Results

### 3.1. Identification of MADS-Box Genes in Theobroma cacao

To identify the MADS-box genes in *Theobroma cacao*, two HMM analyses were performed: after removing duplicates, a total of 68 putative MADS proteins were obtained by first HMMER searches, using the MADS domain profile as a query, in the coco tree protein database. For the second HMM analysis, we selected proteins which e-value > 0.05 as candidate members, choosing the longest transcript for each screened gene, and thus generating 69 *MADS-box* genes after confirming MADS domain by SMART and NCBI Conserved Domain Search Service (Appendix A). These 69 MADS-box genes were sequentially renamed from *TcMADS1* to *TcMADS69* based on their chromosomal location and subjected to further analyses. Detailed characteristics, including number of amino acids (aa), average molecular weight (MW), theoretical pI, instability index, and aliphatic index about *TcMADS* genes, are listed in Table 1. The statistical results showed that the protein length varied, ranging from 78 (*TcMADS23*) to 600 (*TcMADS7*) amino acids, with an average length of amino acids, and the molecular weights varied from 66752.75 Da (*TcMADS23*) to 8995.45 Da (*TcMADS7*). Additionally, thirteen MADS-box proteins were acidic, with pI values less than 6.5; 52 were alkaline, with pI values greater than 7.5; four were neutral, with a pI are between 6.5 and 7.5. The instability index analysis indicated that most of the TcMADS proteins were unstable, with an instability index greater than 40, except for *TcMADS12*, *TcMADS37*, *TcMADS57*, *TcMADS67*, *TcMADS1*, *TcMADS55*, *TcMADS9*, *TcMADS47*. The subcellular localization prediction of *TcMADS* genes was analyzed by BUSCA tools. From the analysis results, most *TcMADS* genes appeared to mainly be located in the nucleus (63.77%) and chloroplast (34.78%), with only *TcMADS11* found in the endomembrane system.

### 3.2. Phylogenetic Analysis and Classification of the MADS-Box Gene

To understand the phylogenetic relationship among MADS-box genes in the coco tree and group them into the established subfamilies, we employed MEGA X to construct a rooted neighbor-joining phylogenetic tree based on the amino acid sequence alignment of 69 proteins from *Theobroma cacao* and 96 from *Arabidopsis* (Figure 1) [15], which also allowed for inferences to be made about the possible function of these genes based on *Arabidopsis* gene function research. According to the general MADS-box gene classification in *Arabidopsis*, the *TcMADS* genes were grouped into two types: type I and type II. Then, based on the phylogenetic relationships, the type I MADS-box genes were further subdivided into more detailed subfamilies: Mα (11), Mβ (2), Mγ (14), Mδ (9). The Mβ group has the minimum number of members, 2, while the corresponding group members in *Arabidopsis* contains 16, which indicates that genes were lost over the development of evolution. and the remaining 32 members were classified as MIKC type II. It is notable that *TcMADS39* is not classified into any of these subfamilies; therefore, we group it as UN.

### 3.3. Conserved Motif and Structure Analysis

To gain insights into the structural diversity and similarity of MADS-box genes in coco tree, we analyzed the intron–exon arrangements and conserved motifs according to their phylogenetic relations. As shown in Figure 2A, we first constructed an individual phylogenetic tree using an NJ method similar to that of the species tree described above, and then mapped their intron–exon structure (Figure 2B). A very striking distribution of introns in the *Arabidopsis* MADS-box genes was previously reported: the MICK subfamily of *TcMADS* genes contained multiple introns, as did the Mδ group, whereas the remaining three subfamilies (Mα, Mβ and Mγ) usually had no introns, or only one or two introns. The reason the Mα, Mβ and Mγ groups contain fewer introns might be a differential tendency to lose or acquire introns or a reverse-transcribed origin for the ancestors of the three subfamilies [18]. In our study, the number of introns in *TcMADS* genes ranged from one (*TcMADS14*, *TcMADS15*, *TcMADS4*, *TcMADS63*, *TcMADS47*, *TcMADS49*) to eighteen (*TcMADS7*). Furthermore, closely related genes have a similar gene structure, differing only in the length of exons and introns. The shortest *TcMADS* gene was just 237 bp in length (*TcMADS23*), while the longest gene was *TcMADS7*, with a length of 1803 bp.

To further study the characteristics of the MADS-box gene family and the conserved motifs that are shared among different subfamilies in *Theobroma cacao*, Multiple Expectation Maximization for Motif Elicitation program was used to identify the conserved motifs. A total of 10 conservative motifs were predicted and named from Motif 1 to Motif 10 (Figure 2C). Among these motifs, Motif 1 was prevalent in all genes; it is worth noting that there were only two Motif 1s in *TcMADS53*. Motif 2 was also present in almost *TcMADS* genes. Motif 3, Motif 8, Motif 5, Motif 9 and Motif 10 were only observed in the Mγ subfamily, which indicated that they might be unique to the Mγ group. Generally, *TcMADS* genes of the same subfamily had similar motifs; we speculated that they might have a similar biological function.

### 3.4. Genome Distribution and Gene Evolution Analysis of TcMADS Genes

According to the location information acquired from genome annotation file downloaded in Ensembl Plants database, 69 *TcMADS* genes were evenly distributed on 10 chromosomals (Figure 3A) and renamed based on their position on the chromosome. A higher abundance of MADS-box genes (18.84%) of coco tree was observed on chromosome (Chr) I and II, whereas ChrVII, ChrXI, ChrX had only two MADS-box genes (2.90%). As shown in Figure 3B, a chromosomal bias was observed in the distribution of Mγ subfamily, which was mainly confined to ChrV. ChrIII and ChrVIII were both contained nine *TcMADS* genes. The other MADS-box genes of *Theobroma cacao* were located as follows: 5, 10 and 4 on ChrIV, ChrV and ChrVI, respectively.

Some of the MADS-box genes distribution showed a relatively high density on chromosomes. We screened tandem duplicated gene pairs among sixty-nine *TcMADS* genes. The analysis showed that three genes (*TcMADS41*, *TcMADS42*, *TcMADS43*) on ChrV are duplications of each other, and two gene pairs were also found on ChrV (*TcMADS49*&*TcMADS50*, *TcMADS45*&*TcMADS46*) and one pair on ChrII (*TcMADS68*&*TcMADS69*). Additionally, the substitution ratio of non-synonymous (Ka) to synonymous (Ks) mutations (Ka/Ks) of above six pairs were calculated. As shown in Table 2, Ka/Ks values of *TcMADS43*&*TcMADS42* and *TcMADS43*&*TcMADS41* > 1, which means that these genes were positively selected over the course of evolution and the new protein functions could be beneficial to the survival and reproduction of the coco tree. The remaining four gene pairs had Ka/Ks < 1, indicating that these duplicated gene pairs evolved under purifying selection. 

### 3.5. Analysis of Putative Promoter Regions in TcMADS Genes

The cis-regulatory elements serve as a molecular switch by binding to transcription factors, which are associated with gene transcription initiation and transcription activity. To explore the putative functions of *TcMADS* genes, we extracted and examined the 2k bp sequences upstream the transcription start site. Four types of cis-acting elements were present in the promoter regions when submitted to PlantCARE Online program, including a light-responsive element, wound-responsive element, gibberellin-responsive element and auxin-responsive element. These were identified in our study, indicating that *TcMADS* genes are closely related to abiotic stress response. The distribution of these cis-acting elements on the promoters is shown in Appendix A. Light-responsive elements were present in almost all promoter regions of MADS-box genes, with an especially large number in *TcMADS28*, *TcMADS46*, *TcMADS62*, *TcMADS65*.

## 4. Discussion

MADS-box proteins are major transcription factors involved in almost every biological process, and a surprising number of them have been systematically identified and analyzed in a variety of species. Although many MADS-box genes have been shown to have conserved functions in flower development and fruit ripening [38,39,40,41], some MADS-box genes have acquired novel functions in specific species during evolution [42]. To date, no detailed analysis of MADS-box genes has been performed in *Theobroma coco*. A better understanding of this family in terms of their member feature, structure characteristics can provide new ideas for further functional analysis. Compared with previous studies, the number of this family member varies in different species, with 107 in *Arabidopsis* [18], 105 in *populus trichocarpa* [19], 48 in *pineapple* [20], 182 in *Saccharum spontaneum* [21], 44 in *Erigeron breviscapus* [19,22], 64 in *Salix suchowensis* [43]. A total of 69 MADS-box proteins were identified from the coco tree in this study (Table 1), which is less than that in *Arabidopsis*. One possible explanation for this is that MADS-box genes coco tree may have a higher gene loss rate compared to that of *Arabidopsis*, indicating an important role of gene duplication over the course of evolution in various species [44]. These 69 *TcMADS* genes were renamed (*TcMADS1-TcMADS69*) based on their chromosomal location and further classified two types according to their phylogenetic relationship with *Arabidopsis*: type I including subclass Mα (11 genes), Mβ (2 genes), Mγ (14 genes), Mδ (9 genes) and type II MIKC (32 genes). The remaining *TcMADS* gene was classified as group UN. We found that most MADS-box genes belong to the MIKC subfamily, and *Theobroma coco* had a comparable number of Mδ and Mγ genes but fewer Mα, Mβ and MIKC genes than *Arabidopsis*, meaning that *Arabidopsis* may undergo more gene duplication events than *Theobroma cacao*. The structures of two types of MADS-box genes were obviously different, and MIKC subgroup genes were more conservative compared with other groups. Additionally, Type II genes usually have multiple introns, whereas most Mα, Mβ and Mγ members have fewer or no introns, indicating that these genes may experience more intron loss during gene family diversification. Previous studies proposed that the number of gene introns correlates with the expression level: the fewer the introns, the higher the pression [45,46]. The same pattern of intron–exon structures in type I and type II exists among diverse species including watermelon [47], *Brachypodium distachyon* [48], rice [49], and lettuce [50]. Overall, genes within the same group are structurally different from other genes; therefore, we speculated that there may be a complicated gene structural evolution in *TcMADS* genes.

Phylogenomic analyses shows that gene and genome duplication events usually contribute to the diversification of the MADS-box transcription factor and play significant roles in shaping the regulatory networks involved in key phenotypic characters [51]. In this study, six tandem-duplicated gene pairs were identified, which all belong to the Mδ subfamily. As ubiquitous genetic components, promoters drive gene transcription and precisely, temporarily and spatially control gene in response to developmental and environmental signals [52]. The cis-acting elements located upstream of the transcription start sites play a vital biological role in regulating gene expression during growth and development [53]. A promoter analysis indicated that *TcMADS* genes are involved in diverse stress and hormone responses, making it possible to study individual gene function.

## 5. Conclusions

In this study, a systematic analysis was conducted of the Theobroma MADS-box gene family. Based on the *Theobroma cacao* genome data, we used HMM profiles to identify 69 MADS-box genes. Sixty-nine MADS-box genes were distributed across 10 chromosomes and phylogenetically classified into six subfamilies, which showed high similarity in terms of gene structure and conserved motifs within the same subfamily. Furthermore, cis-acting elements analysis show that *TcMADS* genes may be involved in diverse stress responses. In summary, these results provided more information about MADS-box genes and establish a foundation for future study of MADS-box genes in *Theobroma cacao*.

## Figures and Tables

**Figure 1 genes-12-01799-f001:**
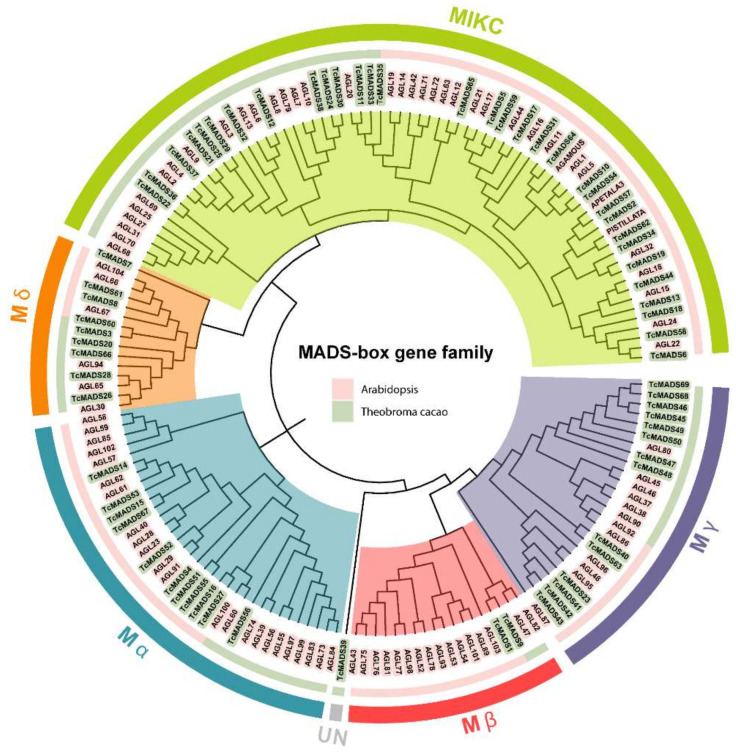
Phylogenetic tree of MADS-box genes in *Arabidopsis* and *Theobroma cacao*. The MADS-box genes are indicated with light pink and light green shade for *Arabidopsis* and *Theobroma cacao,* respectively. In second (narrow) ring from outside, the size of the area represented by the two colors shows the proportion of genes from two species in each group. The subgroups are marked by colorful background and circles.

**Figure 2 genes-12-01799-f002:**
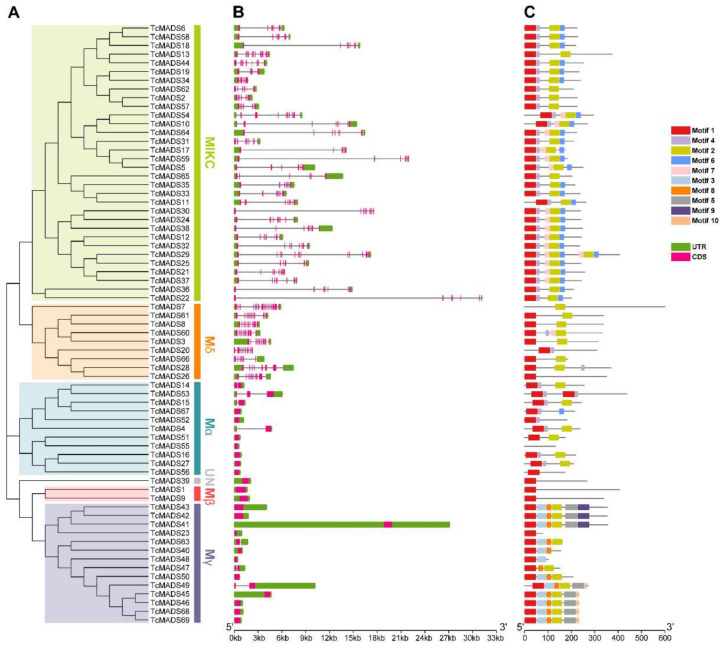
Phylogenetic relationship, gene structure and conserved motifs of the *TcMADS* genes. (**A**) An unrooted NJ tree (left side of the figure) obtained using the MEGA X based on coco tree MADS-box protein sequences. (**B**) The exon–intro structures of *Theobroma cacao* MADS-box genes (central of the figure) were displayed by TBtools software. (**C**) Conserved motif composition of the TcMADS proteins (right side). Detailed information on the ten motifs is provided in Appendix A.

**Figure 3 genes-12-01799-f003:**
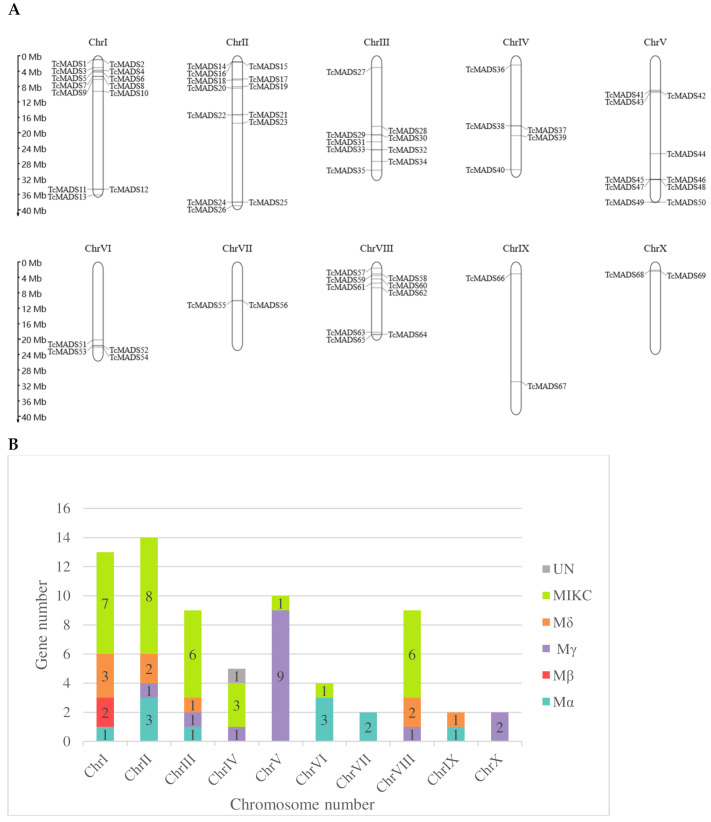
(**A**) Physical distribution of *TcMADS* genes among 10 chromosomes. (**B**) Number of TcMADS subfamily on each chromosome.

**Table 1 genes-12-01799-t001:** Detailed information regarding *MADS-box* gene family in *Theobroma cacao*.

Gene Name	Gene ID	Physicochemical Characteristics					SL	ORF
		PI	MW (Da)	Length (aa)	Instability Index	Aliphatic Index		
*TcMADS1*	TCM_000239	9.48	46,436.9	406	39.09	77.32	nucleus	1221
*TcMADS2*	TCM_000266	9.59	26,095.8	224	46.52	87.1	chloroplast	675
*TcMADS3*	TCM_000725	9.91	36,097.68	315	58.23	83.87	nucleus	948
*TcMADS4*	TCM_000878	6.85	26,140.89	237	41.11	78.23	chloroplast	714
*TcMADS5*	TCM_000931	9.12	29,034.14	250	45.33	84.56	chloroplast	753
*TcMADS6*	TCM_000992	6.55	25,353.79	224	53.32	85.76	nucleus	675
*TcMADS7*	TCM_001181	5.86	66,752.75	600	48.76	77.52	nucleus	1803
*TcMADS8*	TCM_001182	5.43	37,831.38	337	62.2	73.77	nucleus	1014
*TcMADS9*	TCM_001335	9.19	38,279.15	338	38.49	68.11	nucleus	1017
*TcMADS10*	TCM_001841	9.85	31,109.18	269	62.98	6.17	chloroplast	810
*TcMADS11*	TCM_005456	8.91	30,163.08	262	53.63	84.89	endomembrane system	789
*TcMADS12*	TCM_005458	9.08	27,810.84	243	38.49	82.3	nucleus	732
*TcMADS13*	TCM_005818	8.51	42,280.27	375	42.75	101.09	nucleus	1128
*TcMADS14*	TCM_006323	9.42	28,732.02	254	52.22	62.24	nucleus	765
*TcMADS15*	TCM_006324	9.15	27,699.77	243	43.97	85.56	nucleus	732
*TcMADS16*	TCM_006325	5.42	24,228.12	218	45.89	55.96	nucleus	657
*TcMADS17*	TCM_007324	9.52	20,330.45	174	48.32	100.29	chloroplast	525
*TcMADS18*	TCM_007378	8.96	24,754.12	219	51.13	85.11	chloroplast	660
*TcMADS19*	TCM_007713	7.74	27,574.3	233	66.54	80.73	chloroplast	702
*TcMADS20*	TCM_007787	9.12	36,022.57	310	46.31	92.13	nucleus	933
*TcMADS21*	TCM_008703	8.5	29,428.29	258	46.99	82.79	nucleus	777
*TcMADS22*	TCM_008716	5.92	22,921.21	199	56.26	89.65	nucleus	600
*TcMADS23*	TCM_008973	9.39	8995.45	78	51.89	96.15	chloroplast	237
*TcMADS24*	TCM_011475	8.97	28,016.08	241	57.36	80.17	nucleus	726
*TcMADS25*	TCM_011478	6.61	27,447.96	240	58.38	73.62	nucleus	723
*TcMADS26*	TCM_011687	6.33	39,385.48	351	47.73	78.66	nucleus	1056
*TcMADS27*	TCM_012489	6.85	23,710.15	210	40.52	71.57	nucleus	633
*TcMADS28*	TCM_014051	6.13	41,766.39	370	51.46	77.22	chloroplast	1113
*TcMADS29*	TCM_014337	8.79	46,247.61	407	52.14	88.87	nucleus	1224
*TcMADS30*	TCM_014345	9.06	27,737.47	239	50.17	76.78	nucleus	720
*TcMADS31*	TCM_014661	9.83	24,429.07	210	55.07	86.33	chloroplast	633
*TcMADS32*	TCM_015044	8.82	27,249.02	236	53.76	86.78	nucleus	711
*TcMADS33*	TCM_015049	9.88	27,106.38	237	47.23	90.13	chloroplast	714
*TcMADS34*	TCM_015674	5.47	27,657.45	238	69.7	87.65	nucleus	717
*TcMADS35*	TCM_016147	9.24	24,830.39	215	63.31	73.95	nucleus	648
*TcMADS36*	TCM_017242	8.51	24,320.82	209	47.08	86.75	nucleus	630
*TcMADS37*	TCM_018979	9.07	27,740.63	243	34.06	85.14	nucleus	732
*TcMADS38*	TCM_018981	8.77	28,366.14	248	62.86	80.24	nucleus	747
*TcMADS39*	TCM_019362	8.23	30,811.58	267	58.24	70.15	nucleus	804
*TcMADS40*	TCM_021050	9.7	17,902.07	155	45.29	89.94	chloroplast	468
*TcMADS41*	TCM_022993	9.51	40,637.63	356	53.16	76.71	chloroplast	1071
*TcMADS42*	TCM_023006	9.2	38,815.19	354	58.36	70.54	nucleus	1065
*TcMADS43*	TCM_023041	8.93	38,451.37	354	59.01	69.49	nucleus	1065
*TcMADS44*	TCM_024579	6.04	28,901.4	252	61.39	85.48	nucleus	759
*TcMADS45*	TCM_025670	8.86	26,594.65	233	64.66	74.12	chloroplast	702
*TcMADS46*	TCM_025671	9.37	26,384.43	233	63.02	70.39	nucleus	702
*TcMADS47*	TCM_025674	9.64	16,948.73	150	36.03	79.4	nucleus	453
*TcMADS48*	TCM_025676	10.29	11,744.9	103	56.26	68.25	chloroplast	312
*TcMADS49*	TCM_026842	9.26	30,499	273	46.06	71.87	nucleus	822
*TcMADS50*	TCM_026845	9.47	23,787.52	207	49.15	78.74	chloroplast	624
*TcMADS51*	TCM_029234	4.91	19,812.05	174	42.71	76.21	nucleus	525
*TcMADS52*	TCM_029518	9.52	20,156.2	182	44.45	79.84	chloroplast	549
*TcMADS53*	TCM_029519	9.64	49,499.24	437	50.15	68.56	nucleus	1314
*TcMADS54*	TCM_029596	9.68	34,012.76	294	66.03	74.05	nucleus	885
*TcMADS55*	TCM_032402	9.25	13,764.45	132	34.38	60.68	nucleus	399
*TcMADS56*	TCM_032403	7.74	19,620.34	172	50.98	74.24	chloroplast	519
*TcMADS57*	TCM_034148	9.08	26,051.7	225	33.21	88.36	chloroplast	678
*TcMADS58*	TCM_034501	7.64	25,504.16	227	55.81	91.06	nucleus	684
*TcMADS59*	TCM_034549	9.62	21,512.84	184	50.3	88.42	chloroplast	555
*TcMADS60*	TCM_034757	5.45	37,593.54	333	59.52	84.83	nucleus	1002
*TcMADS61*	TCM_034970	5.26	38,476.16	337	60.18	82.43	nucleus	1014
*TcMADS62*	TCM_035212	8.87	24,432.87	209	61.75	78.42	nucleus	630
*TcMADS63*	TCM_036473	9.34	18,742.04	162	41.19	99.38	chloroplast	489
*TcMADS64*	TCM_036541	9.43	25,550.24	222	61.61	91.4	chloroplast	669
*TcMADS65*	TCM_036568	9.52	23,170.78	203	56.62	87	nucleus	612
*TcMADS66*	TCM_037394	9.62	21,551.01	186	42.9	92.31	chloroplast	561
*TcMADS67*	TCM_040735	9.76	24,088.7	214	39.18	87.01	chloroplast	645
*TcMADS68*	TCM_042799	4.84	26,046.39	233	57.5	68.28	nucleus	702
*TcMADS69*	TCM_042848	4.81	26,121.69	233	55.41	75.41	nucleus	702

**Table 2 genes-12-01799-t002:** Tandem duplicated gene pairs and their Ka, Ks, Ka/Ks values.

Tandem Duplicated Gene Pairs	Chromosome	Ka	Ks	Ka/Ks
*TcMADS43*&*TcMADS42*	ChrV	0.108301	0.103758	1.04379
*TcMADS43*&*TcMADS41*	ChrV	0.213254	0.170408	1.25143
*TcMADS42*&*TcMADS41*	ChrV	0.215878	0.236528	0.912695
*TcMADS49*&*TcMADS50*	ChrV	0.098915	0.237876	0.415826
*TcMADS45*&*TcMADS46*	ChrV	0.078058	0.111367	0.700902
*TcMADS68*&*TcMADS69*	ChrII	0.035449	0.086453	0.410042

## Data Availability

Data will be available on reasonable request.

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
