# Peer review of "Genome-Wide Identification and Analysis of the MADS-Box Gene Family in *Theobroma cacao"

_genes, 2021, doi:10.3390/genes12111799_

Round 1

Reviewer 1 Report

The manuscript entitled “Genome-Wide Identification and Analysis of the MADS-Box Gene Family in Theobroma cacao” has conducted a systematic analysis of Theobroma MADS-box gene family. The authors have conducted a detailed analysis for MADS-Box which would be helpful for the researcher of this area. However, my major concern is the grammatical and linguistic errors in the manuscript. I would like to request the authors to improve the language and typos in the entire manuscript. I have pointed out some of them, but there are many more that need to be corrected to improve the readability of the manuscript. Line no 69 replace “perfumed” with “performed” In line no 52 rewrite the sentence “there has been a great deal research on the MADS-box gene family”. In Line no 59 please replace “Additionally, there were studies proposed that” with “Additionally, some studies proposed that ”. In line no 248 please remove the comma from “The cis-regulatory elements, serves as a molecular switch”. In line no 267, please rewrite the sentence “There are 14 genes expressed in at least on one type of coco pod”. In line no 292 please replace “an important roles” with “an important role”. In the conclusion section please remove line no 329. Additionally, I would also like to suggest authors to perform qRT-analysis of a few highly expressed genes to validate the expression patterns of Theobroma cacao.

Author Response

Response to Reviewer 1 Comments

Dear teacher,

 Thank you for taking your time process the submission of our original paper entitled “Genome-Wide Identification and Analysis of the MADS-Box Gene Family in Theobroma cacao”.

We appreciate your kind suggestions and we have carefully amended the manuscript accordingly. We sincerely apologize for the inconvenience of reading caused by grammatical problems and typos. Accordingly, we have corrected the problems and all similar ones throughout the manuscript without altering the paper’s original meaning. To help you evaluate our revised manuscript at ease, I wish to provide a point-by-point response to the comment as follows:

Point 1: Line no 69 replace “perfumed” with “performed”.

Response 1: Thank you. The following is the revised sentence,

subcellular localization and cis-acting elements were also performed.

Point 2: In line no 52 rewrite the sentence “there has been a great deal research on the MADS-box gene family”.

Response 2: Thank you. The following is the revised sentence,

the MADS-box proteins have been characterized in various kinds of plants,

Point 3: In Line no 59 please replace “Additionally, there were studies proposed that” with “Additionally, some studies proposed that”.

Response 3: Thank you. The following is the revised sentence,

some studies have proposed that an ingredient found in coco might exert cardiovascular benefits.

Point 4: In line no 248 please remove the comma from “The cis-regulatory elements, serves as a molecular switch”. 

Response 4: Thank you. The following is the revised sentence,

The cis-regulatory elements serve as a molecular switch by binding to transcription factors, which are associated with gene transcription initiation and transcription activity.

Point 5: In line no 267, please rewrite the sentence “There are 14 genes expressed in at least on one type of coco pod”.

Response 5: Thank you for your constructive comments. Considering the accuracy of conclusion and other reviewer’s suggestions, we deleted the part of gene expression analysis.

Point 6: In line no 292 please replace “an important roles” with “an important role”.

Response 6: Thank you. The following is the revised sentence,

indicating an important role of gene duplication over the course of evolution in various species,

Point 7: In the conclusion section please remove line no 329.

Response 7: Thank you. The line no 329 has been removed.

Point 8: Additionally, I would also like to suggest authors to perform qRT-analysis of a few highly expressed genes to validate the expression patterns of Theobroma cacao.

Response 8: Thank you for your constructive comments. Considering the accuracy of our conclusion and other reviewer’s suggestions, we deleted the part of gene expression analysis here. We believe our results will be more critical with qRT-analysis and we will follow your advice and do further expression pattern analysis in the follow-up studies.

All changes made are highlighted in red.

We hereby resubmit the revised manuscript and hope that all corrections are satisfactory. Please feel free to contact us with any questions and we look forward to your decision.

Yours sincerely,

Qianqian Zhang

Reviewer 2 Report

Review for the manuscript „Genome-wide identification and analysis of the MADS-Box Gene Family in Theobroma cacao“ by Zhang et al.

Authors presented the identification, classification and partial characterization of 69 T. cacao genes with significant homology to Arabidopsis thaliana MADS-box genes. For economically important cacao tree this can be the starting point for further research on regulation of physiological and developmental processes in flower-, fruit- and seed development. The manuscript is generally well written, but some questions remain open:

  • Line 143-166: In this section is described once more the content of table 1. Description of the minimal and maximal values is only necessary for the most important characteristics, not for all. All other data can be found in the table.
  • Line 154: Are you sure that the TcMADS23 protein is complete?
  • Figure 1: The second (narrow) ring from outside (giving information about the relation of known genes in thaliana and T. cacao) is not only unnecessary because inside the genes are color-coded, it makes confusions. So I would suggest to delete it from the figure.
  • Line 221: You wrote „… were renamed based on their position on the chromosome“. This causes at that position of the manuscript confusions. Somebody can ask, if the names of the genes presented in tab 1., fig. 1 and fig. 2 are now not consistant with the names of the genes in fig.3 ff. This should absolutely be avoided.
  • Figure 3A: The names of the genes are not easily readable in the printed version. Because the readers need them in the following discussion on duplicated genes, this should be greater!
  • Line 236-237 and ff: TcMADS41, TcMADS42 and TcMADS43 are duplications of each other, so these are not 3 duplications but 1 triplication.
  • Line 259-277 (most important point of criticism): You were not able to show significant impact of expression pattern of any of the 14 MADS-box genes on coco fruit color phenotype (Figure 4). The number of the tested plants (3x GF, 3x GW, 3x PF) is by far too small and there is no clear tendency for one of these genes. I am sure that such differences in expression level you will find for hundreds of cacao genes. So I would suggest to delete this point completely from introduction, M&M (2.6), results (Expression analysis of MADS-box genes in Theobroma cacao, 3.5, but must be 3.6), discussion and conclusion. In the current state of your investigation this is completely speculative.
  • Line 329-330: What information should give the first sentence of conclusion?
  • Complete text: Genes should be written in kursiv, but proteins not.
  • Smaller mistakes: for instance: lines 79, 248, 250.

Author Response

Response to Reviewer 2 Comments

Dear teacher,

 Thank you for taking your time process the submission of our original paper entitled “Genome-Wide Identification and Analysis of the MADS-Box Gene Family in Theobroma cacao”.

We appreciate your kind suggestions and we have carefully amended the manuscript accordingly. To help you evaluate our revised manuscript at ease, I wish to provide a point-by-point response to the comment as follows:

Point 1: Line 143-166: In this section is described once more the content of table 1. Description of the minimal and maximal values is only necessary for the most important characteristics, not for all. All other data can be found in the table.

Response 1: We have appropriately removed descriptions about some of the features in Table 1 and kept only some most important characteristics to make the article concise and clear. The following is the revised sentence,

“Detailed characteristics, including number of amino acids (aa), average molecular weight (MW), theoretical pI, instability index, and aliphatic index about TcMADS genes, are listed in Table 1. The statistical results showed that the protein length varied, ranging from 78 (TcMADS23) to 600 (TcMADS7) amino acids, with an average length of amino acids, and the molecular weights varied from 66752.75 Da (TcMADS23) to 8995.45 Da (TcMADS7). Additionally, thirteen MADS-box proteins were acidic, with pI values less than 6.5; 52 were alkaline, with pI values greater than 7.5; four were neutral, with a pI are between 6.5 and 7.5. The instability index analysis indicated that most of the TcMADS proteins were unstable, with an instability index greater than 40, except for TcMADS12, TcMADS37, TcMADS57, TcMADS67, TcMADS1, TcMADS55, TcMADS9, TcMADS47. The subcellular localization prediction of TcMADS genes was analyzed by BUSCA tools. From the analysis results, most TcMADS genes appeared to mainly be located in the nucleus (63.77%) and chloroplast (34.78%), with only TcMADS11 found in the endomembrane system.”

Point 2: Line 154: Are you sure that the TcMADS23 protein is complete?

Response 2: We searched the protein sequence of TcMADS23 on the website (Ensembl Plants) and found that it was complete.

Point 3: Figure 1: The second (narrow) ring from outside (giving information about the relation of known genes in thaliana and T. cacao) is not only unnecessary because inside the genes are color-coded, it makes confusions. So I would suggest to delete it from the figure.

Response 3: Thank you for your constructive comments and we understand this point. We apologize for any confusion caused and appreciate the valuable suggestions. In second (narrow) ring from outside, the size of the two color areas represents the approximate proportion of genes from two species in each group. For example, the larger the pink area, the larger the number of genes in Arabidopsis is in that group. I should have explained that in the caption.

Point 4: Line 221: You wrote “… were renamed based on their position on the chromosome”. This causes at that position of the manuscript confusions. Somebody can ask, if the names of the genes presented in tab 1., fig. 1 and fig. 2 are now not consistent with the names of the genes in fig.3 ff. This should absolutely be avoided.

Response 4: Thank you for your comments and we understand this point, we first analysed their position on the chromosome, and then renamed these genes based on their chromosomal location. Finally, we summarized the final names in Table 1 when we wrote this article, so the names of the genes presented in tab 1., fig. 1 and fig. 2 are consistent with the names of the genes in fig.3 and other figures and tables.

Point 5: Figure 3A: The names of the genes are not easily readable in the printed version. Because the readers need them in the following discussion on duplicated genes, this should be greater!

Response 5: Thank you. I have redrawn Figure 3A.

Point 6: Line 236-237 and ff: TcMADS41, TcMADS42 and TcMADS43 are duplications of each other, so these are not 3 duplications but 1 triplication.

Response 6: Thank you. The following is the revised sentence,

“Some of the MADS-box genes distribution showed a relatively high density on chromosomes. We screened tandem duplicated gene pairs among sixty-nine TcMADS genes. The analysis showed that three genes (TcMADS41, TcMADS42, TcMADS43) on ChrV are duplications of each other, and two gene pairs were also found on ChrV (TcMADS49&TcMADS50, TcMADS45&TcMADS46) and one pair on ChrII (TcMADS68&TcMADS69)”

Point 7: Line 259-277 (most important point of criticism): You were not able to show significant impact of expression pattern of any of the 14 MADS-box genes on coco fruit color phenotype (Figure 4). The number of the tested plants (3x GF, 3x GW, 3x PF) is by far too small and there is no clear tendency for one of these genes. I am sure that such differences in expression level you will find for hundreds of cacao genes. So I would suggest to delete this point completely from introduction, M&M (2.6), results (Expression analysis of MADS-box genes in Theobroma cacao, 3.5, but must be 3.6), discussion and conclusion. In the current state of your investigation this is completely speculative.

Response 7: We would like to express our sincere thanks for your comments. Considering the rationality of your suggestions and the incompleteness of our conclusions, we have decided to delete this content according to your suggestions. Relevant parts of the full text have been deleted.

Point 8: Line 329-330: What information should give the first sentence of conclusion?

Response 8: Thank you. The first sentence of conclusion is the sentence in file “genes-template.dot”. I am so sorry that I forgot to delete it when I changed the format.

Point 9: Complete text: Genes should be written in kursiv, but proteins not.

Response 9: Thank you. I have checked the full text and corrected similar problems.

Point 10: Smaller mistakes: for instance: lines 79, 248, 250.

Response 10: Thank you. I have checked the full text and corrected similar mistakes.

Line79: MADS-box proteins in Theobroma cacao were searched using the following two approaches.

Line248: The cis-regulatory elements serve as a molecular switch by binding to transcription factors, which are associated with gene transcription initiation and transcription activity.

Line250: To explore the putative functions of TcMADS genes, we extracted and examined the 2k bp sequences upstream the transcription start site.

All changes made are highlighted in red.

We hereby resubmit the revised manuscript and hope that all corrections are satisfactory. Please feel free to contact us with any questions and we look forward to your decision.

Yours sincerely,

Qianqian Zhang
